# Ginger Root Extract Improves GI Health in Diabetic Rats by Improving Intestinal Integrity and Mitochondrial Function

**DOI:** 10.3390/nu14204384

**Published:** 2022-10-19

**Authors:** Rui Wang, Julianna Maria Santos, Jannette M. Dufour, Emily R. Stephens, Jonathan M. Miranda, Rachel L. Washburn, Taylor Hibler, Gurvinder Kaur, Dingbo Lin, Chwan-Li Shen

**Affiliations:** 1Department of Pathology, Texas Tech University Health Sciences Center, Lubbock, TX 79430, USA; 2Department of Cell Biology and Biochemistry, Texas Tech University Health Sciences Center, Lubbock, TX 79430, USA; 3Center of Excellence for Integrative Health, Texas Tech University Health Sciences Center, Lubbock, TX 79430, USA; 4Department of Medical Education, Texas Tech University Health Sciences Center, Lubbock, TX 79430, USA; 5Obesity Research Institute, Texas Tech University, Lubbock, TX 79401, USA; 6Department of Nutritional Sciences, Oklahoma State University, Stillwater, OK 74078, USA

**Keywords:** ginger root extract, diabetes, intestinal health, mitochondria function

## Abstract

Background Emerging research suggests hyperglycemia can increase intestinal permeability. Ginger and its bioactive compounds have been reported to benefit diabetic animals due to their anti-inflammatory and antioxidant properties. In this study, we revealed the beneficial effect of gingerol-enriched ginger (GEG) on intestinal health (i.e., barrier function, mitochondrial function, and anti-inflammation) in diabetic rats. Methods Thirty-three male Sprague Dawley rats were assigned to three groups: low-fat diet (control group), high-fat-diet (HFD) + streptozotocin (single low dose 35 mg/kg body weight (BW) after 2 weeks of HFD feeding) (DM group), and HFD + streptozotocin + 0.75% GEG in diet (GEG group) for 42 days. Glucose tolerance tests (GTT) and insulin tolerance tests (ITT) were conducted at baseline and prior to sample collection. Total pancreatic insulin content was determined by ELISA. Total RNA of intestinal tissues was extracted for mRNA expression using qRT-PCR. Results Compared to the DM group, the GEG group had improved glucose tolerance and increased pancreatic insulin content. Compared to those without GEG (DM group), GEG supplementation (GEG group) increased the gene expression of tight junction (Claudin-3) and antioxidant capacity (SOD1), while it decreased the gene expression for mitochondrial fusion (MFN1), fission (FIS1), biogenesis (PGC-1α, TFAM), mitophagy (LC3B, P62, PINK1), and inflammation (NF-κB). Conclusions Ginger root extract improved glucose homeostasis in diabetic rats, in part, via improving intestinal integrity and mitochondrial dysfunction of GI health.

## 1. Introduction

Type 2 diabetes mellitus (T2DM), mainly characterized by hyperglycemia and insulin resistance, is the fastest-growing metabolic disease in the world [1,2,3]. Accumulating evidence has highlighted a strong correlation between T2DM, intestinal barrier dysfunction, oxidative stress, and mitochondrial dysfunction [1,2,3]. Among these, hyperglycemia, insulin resistance, and insulin damage caused by inflammatory cytokines and oxidative stress are the primary reasons [4].

Intestinal barrier integrity is essential for the maintenance of normal intestinal homeostasis and efficient protective reactions against chemical and microbial challenges [5]. The consequences of intestinal barrier defects/dysfunction include (i) increased intestinal permeability, (ii) increased influx of luminal stressors, such as pathogens, toxins, and allergens, and (iii) increased inflammation triggering an immune response [5]. Hyperglycemia drives intestinal barrier disruption causing susceptibility to enteric infection [4]. For example, in a streptozotocin (STZ)-treated mouse model, the diabetic mice developed *C. rodentium* infections and systemic translocation, accompanied by enhanced bacterial growth, epithelial adherence, and systemic spread. STZ treatment also resulted in the dysfunction of intestinal epithelial adherence junctions, coupled with systemic dissemination of microbial products, and enhanced trans-epithelial flux [4].

Mitochondria play a key role in maintaining cellular metabolic homeostasis. Mitochondrial dynamics are controlled by fusion and fission in order to maintain mitochondrial morphology. Mitochondrial fusion is modulated by different proteins, including mitofusin-1 (MFN1), mitofusin-2 (MFN2), and optic atrophy (OPA-1), while mitochondrial fission is controlled by mitochondrial fission 1 (FIS1), dynamin-related protein 1 (DRP1), and mitochondrial fission factor (MFF). PARKIN and (PTEN)-induced putative kinase 1 (PINK1) participate in the process of mitophagy, for which mitochondrial fission is necessary [2]. Excessive generation of mitochondrial reactive oxygen species (ROS) and mitochondrial dysfunction are positively associated with the initiation of inflammation and the development of insulin resistance in T2DM progression [6,7]. In other words, the increase in ROS production with diabetes is associated with alteration in both mitochondrial morphology and redox systems biology [1]. Shan et al. reported that mitochondrial oxidative stress is a major contributor to mitochondrial dysfunction across organ systems, such as the liver, skeletal muscle, and pancreas in T2DM patients [8]. Under hyperglycemic conditions, oxidative stress damages mitochondria, and mitophagy is activated to remove damaged mitochondria. Mitochondria encapsulated in the autophagosome fuse with lysosomes and form the autolysosome to degrade the damaged mitochondria via acidic lysosomal hydrolase [8]. Although the specific pathophysiology observed in each T2DM tissue type is unique due to tissue-specific gene expression patterns influenced by the diabetic state, many cellular processes closely connect to the mitochondrion. As mitochondrial dysfunction is a common cellular pathology associated with T2DM, consideration of novel therapies, such as dietary bioactive compound usage, that improve mitochondrial function to treat T2DM and associated comorbidities would be worthy of further investigation. 

Emerging evidence shows dietary bioactive compounds modulate gut health, intestinal barrier function [9,10], and mitochondrial activity, representing a novel option to modulate energy expenditure and energetic metabolism in cells and tissues [5,11]. Gingerol-enriched-ginger (GEG) with antioxidant and anti-inflammatory properties has been reported to improve glucose metabolism and enhance insulin sensitivity in diabetic animals [12,13,14,15]. However, no studies have investigated how GEG supplementation affects GI health in diabetic rats in terms of intestinal barrier function/integrity, intestinal mitochondrion (fusion, fission, biogenesis, transcription factor, function, and mitophagy), or intestinal oxidative stress/inflammation. Therefore, this study was designed to investigate the effects of dietary GEG supplementation on the intestinal tight junction protein (an indicator of intestinal barrier function), intestinal mitochondrion-associated parameters, and intestinal oxidative stress/inflammation in high-fat diet (HFD), STZ-induced diabetic rats. In the present study, GEG’s effects were investigated in both small (duodenum, jejunum, and ileum) and large (cecum and colon) intestines. We also evaluated the effect of GEG supplementation on pancreatic insulin production and islet morphology in diabetic rats. We hypothesized that GEG supplementation in the diet would improve beta cell function and glucose homeostasis, improve intestinal integrity (a decrease in gene and protein expression of tight junction markers), and mitigate T2DM-induced mitochondrial dysfunction in the intestines. Such beneficial effects from GEG on the GI heath of diabetic rats, in part, would be mediated via suppression of intestinal oxidative stress and inflammation. 

## 2. Materials and Methods

### 2.1. Animals and Treatments

Thirty-three male Sprague Dawley rats (150–180 g body weight) were purchased from Envigo (Cumberland, VA, USA) and housed individually under a 12-h light-dark cycle with food and water *ad libitum*. All procedures were approved by the Institutional Animal Care and Use Committee at Texas Tech University Health Sciences Center (IACUC # 19175). All experiments were performed in accordance with the relevant guidelines and regulations. 

After 5-day acclimation, the rats were randomly stratified by weight and assigned to 3 groups (*n* = 11/group): a low-fat diet group (the control group), an HFD+STZ group (the DM group), and an HFD+STZ+GEG (0.75% *wt*/*wt* GEG in diet) group (the GEG group) for 8 weeks. A single STZ dose (35 mg/kg BW) was administered to both DM and GEG groups after two weeks of an HFD. Throughout the study, the animals in the DM and GEG groups were fed an HFD consisting of 20%, 22%, and 58% of energy from carbohydrates, protein, and fat, respectively, and fat mainly from lard (catalog # D12492, Research Diets Inc., New Brunswick, NJ, USA). Animals had free access to water and food during the study period. Based on the results of gas chromatography-mass spectrometry, GEG consists of 18.7% 6-gingerol, 1.81% 8-gingerol, 2.86% 10-gingerol, 3.09% 6-shogoal, 0.39% 8-shogaol, and 0.41% 10-shogaol [10]. GEG was a gift obtained from Sabinsa Corporation, East Windsor, NJ. Body weight, food intake, and water consumption were recorded weekly. At doses in the range of 100 to 500 mg/kg, GEG has been shown to ameliorate diabetes complications in rats in various diabetic models [16,17,18,19]. Based on these studies, we selected a dose of 0.75% (weight/weight in diet) for our study in HFD+STZ-treated rats, which corresponds to ~300 mg/kg body weight for rats. Figure 1 illustrates experimental design. 

### 2.2. In Vivo Glucose and Insulin Tolerance Tests 

At baseline (time zero, beginning of the study before the group’s assignments and treatments) and at the end of the study (after 6 weeks of GEG intervention), rats fasted for 4 h, and glucose tolerance tests (GTT) were performed by intraperitoneal injection of 2 mg/g body weight of glucose. Blood glucose levels were measured 0, 15, 30, 60, and 120 min following glucose injection. Additionally, insulin tolerance tests (ITT) were performed at baseline and at the end of the study on rats that were fasted for 4 h prior to intraperitoneal injection of 1 U/kg body weight of insulin (Humulin, Abbott, Chicago, IL, USA). Blood glucose levels were analyzed at 0, 15, 30, 60, and 120 min following insulin injection. The total area under the curve (AUC) for both GTT and ITT was calculated by the trapezoidal method. For both GTT and ITT, blood was collected from the tail vein and measured using a glucometer (AmiStrip Plus Blood Glucose Meter, Germaine Laboratories, Inc., San Antonio, TX, USA).

### 2.3. Sample Collection 

At the end of the experiment, animals fasted for 4 h before sample collection. The animals were anesthetized with isoflurane and euthanized, and their blood was drawn for plasma and serum collection. The pancreases were fixed in Z-fix for immunohistochemistry (IHC) or frozen at −80 °C in acetic acid for insulin hormone extraction [20]. The colons were collected and fixed in 4% paraformaldehyde in 1× PBS, transferred to 30% sucrose with 1× PBS, and embedded in OCT for subsequent frozen sectioning and IHC. The intestinal tissues from the studied rats (duodenum, jejunum, ileum, cecum, and colon) were collected, immersed in liquid nitrogen, and stored at −80 °C for later analysis. 

### 2.4. Insulin Measurement 

Pancreases (Control *n* = 8, DM *n* = 8, and GEG *n* = 6) were collected at the end of the study and cellular insulin content was determined by acetic acid extraction followed by mouse insulin ELISA (EMD Millipore Co., Billerica, MA, USA). 

### 2.5. Analysis of Pancreatic Tissue 

At the end of the study, pancreases (*n* = 3) were collected for histological assessment. Tissue was fixed in Z-fix, embedded in paraffin, and tissue sections were immunostained as described previously [20]. Primary antibodies were guinea pig anti-insulin (diluted 1:1000; Dako Agilent Pathology Solutions, Santa Clara, CA, USA) and mouse anti-glucagon (diluted 1:5000; Sigma). Appropriate biotinylated secondary antibodies and avidin–biotin–enzyme complexes were purchased from Vector Laboratories (Burlingame, CA, USA). Diaminobenzidine as the chromogen was purchased from BioGenex (Fermont, CA, USA). Tissue sections were counterstained with hematoxylin. 

### 2.6. RNA Isolation and qRT-PCR 

Total RNA was isolated from intestinal tissues (namely, duodenum, jejunum, ileum, cecum, and colon) using the RNAzol RT (RN190, Molecular Research Center Inc., Cincinnati, OH, USA), BAN ratio 1:200 (BN191, Molecular Research Center, Cincinnati, OH, USA). Total RNA was quantified using nanodrop at 260 nm, (Nanodrop one, Thermo Scientific, Waltham, MA, USA) and then reverse transcribed into cDNA using Maxima first strand cDNA synthesis kit synthesis with dsDNase (Thermo Scientific, K1672, Waltham, MA, USA) using the thermal cycler Bio-rad S1000 (Bio-Rad Laboratories, Inc., Hercules, CA, USA). qRT-PCR was performed on Quant Studio 12K Flex real-time PCR system (Life Technologies, 4470689, Carlsbad, CA, USA) using samples of cDNA for amplification of target genes with β-actin as the control with Universal SYBR green supermix (Bio-rad Laboratories, Inc., 17251-24, Hercules, CA, USA). Table 1 lists the genes tested including Claudin-3, MFN1, FIS1, PGC-1α, TFAM, PINK1, P62, LC3B, SOD1, NF-kB, and β-actin. All gene expressions were normalized to our control β-actin. Gene expression was calculated by the following formula: 2-(ΔCT * 1000) [21].

### 2.7. Immunohistochemistry for Claudin-3 and PINK1 in the Colon 

The OCT-embedded colon tissues were sectioned using a Cryostat (Thermo Fisher Shandon Cryotome E) (10 µm), and the sections were stored in 1× PBS at 4 °C before staining. Sections were washed in 1× PBS (3 times × 10 min) and then permeabilized for 10 min in 1× PBS containing 0.3% Triton-x. Sections were washed in 1× PBS (3 × 5 min), incubated with a blocking solution containing 10% goat serum in PBS for 1 h, and washed in 1× PBS (3 × 5 min). Sections were incubated overnight 4 °C in the primary antibody, PINK1 antibody (Novus Biologicals, BC100–494, dilution 1:200) or Claudin-3 antibody (Invitrogen, 34–1700, dilution 1:100) in 1× PBS containing 1% goat serum. On the next day, sections were washed in PBS (3 × 5 min) and then incubated with secondary antibody (Invitrogen, Goat anti-rabbit IgG conjugated to Alexa Fluor Plus 594, A32740, dilution 1: 2000) for 1 h in darkness. Sections were rinsed with 1× PBS (3 × 5 min), cover-slipped with DAPI mounting solution, and stored at 4 °C. Images of the sections were acquired using the Olympus Confocal Microscope (Fluoview FV2000) at 60×. Fluorescence mean intensity was quantified by selecting a specific ROI for each cell using the FIJI software (ImageJ).

### 2.8. Statistical Analysis 

The data is presented as a mean ± standard error of the mean (SEM) and analyzed by one-way by *post hoc* Tukey’s test with GraphPad Prism 9 (GraphPad Software, San Diego, CA, USA). A significance level of *p*-value < 0.05 applies to all statistical tests. Additionally, we also presented the comparisons with 0.05 < *p* < 0.1 to show a tendency. For GTT, ITT, and AUC analysis, data is presented as mean with error bars or ribbons indicating a 90% confidence interval. One-way ANOVA was used, followed by Tukey’s HSD in RStudio Version 1.4.1106 (RStudio, Boston, MA, USA).

## 3. Results

### 3.1. Glucose Homeostasis: Glucose Tolerance Test (GTT) and Insulin Tolerance Test (ITT)

As expected, there were no statistical differences in the GTT and ITT results between the three treatment groups at the baseline (*p* > 0.05, data not shown). At the end of the study, the rats in the DM group had significantly impaired glucose tolerance (Figure 2A) compared to those in the control group, characteristic of diabetes. For GTT, GEG supplementation (the GEG group) significantly improved glucose tolerance compared to rats without GEG supplementation (the DM group) (Figure 2A). More specifically, after glucose injection, GEG supplemented HFD-STZ rats had significantly lower blood glucose levels (Figure 2A) and overall improved glucose tolerance (indicated by a decreased GTT AUC (Figure 2B) as compared to those without GEG supplementation (the DM group). Regarding ITT results, (i) before injection of insulin, the DM and GEG groups had significantly higher blood glucose levels than the control group; and (ii) there was no significant difference in the response after insulin administration between the DM group and GEG group, as shown by blood glucose levels (Figure 2C) and AUC results (Figure 2D). 

### 3.2. Insulin and Histological Assessment of Pancreatic Tissue

The DM group had significantly lower total pancreatic insulin content compared to that of the control group (Figure 3). Dietary supplementation with GEG resulted in significantly increased levels of insulin within the pancreas of diabetic rats. The order of pancreatic insulin content was Control group > GEG group > DM group (Figure 3, *n* = 6–8 per group).

The increased pancreatic insulin detected in the GEG group is reflected in the increase in insulin-producing beta cells observed throughout the islets as determined by insulin IHC (Figure 4A–C). Histological analysis of the alpha cells in the pancreas revealed a normal distribution of glucagon-producing alpha cells along the outer edge of the islets in the Control group (Figure 4D). On the other hand, in the DM group, the alpha cells were disorganized and located throughout the islet (Figure 4E). For the GEG group, alpha cells were also located throughout the islet intermixed with the beta cells (Figure 4C,F). 

### 3.3. mRNA Expression of Intestinal Tight Junction Marker Assessment

Effects of GEG supplementation on the mRNA expression of tight junction markers, namely Claudin-3, were assessed on duodenum, jejunum, ileum, cecum, and colon (Figure 5A). Compared to the control group, the DM group showed a decrease in Claudin-3 mRNA levels in the studied tissues. Compared to the DM group, the GEG group had increased Claudin-3 mRNA expression in the ileum and colon. Figure 5B shows the representative Claudin-3 protein expression of three respective groups using IHC. The DM group had significantly lower Claudin-3 protein expression compared to the control group, and the GEG group had significantly higher Claudin-3 protein expression than the DM group in the colon and ileum.

### 3.4. mRNA Expression of the Intestinal Mitochondrial Fusion, Fission, and Biogenesis Markers

We investigated GEG supplementation on MFN1 and FIS1 gene expression for mitochondrial fusion and mitochondrial fission, respectively, in the small intestine (duodenum, jejunum, and ileum) and the large intestine (cecum and colon). Compared to the control group, the DM group had greater levels of MFN1 gene expression in the duodenum, jejunum, ileum, cecum, and colon (Figure 6). The GEG administration resulted in lower levels of MFN1 gene expression in all studied intestinal tissues in HFD+STZ-treated rats. 

In terms of the FIS1 gene, the FIS1 gene expression levels were elevated in the jejunum, ileum, cecum, and colon of HFD+STZ-treated animals (DM group), as compared to the control group (Figure 7). The GEG group tended to suppress FIS1 gene expression in the ileum (Figure 7, 0.05 < *p* < 0.1). 

We evaluated the effects of GEG supplementation on PGC-1α gene expression, a multi-functional transcriptional coactivator involved in mitochondrial biogenesis, in DM rats (Figure 8). The gene expression levels of PGC-1α were higher in the duodenum, jejunum, ileum, and cecum of the DM groups than those in the control group (Figure 8). GEG supplementation in the diet suppressed the T2DM-induced PGC-1α gene expression in the duodenum, jejunum, ileum, and cecum (Figure 8). We examined GEG’s effects on gene expression of TFAM, a transcription factor A required for mitochondrial biogenesis (Figure 9). Compared to the control group, the DM group had greater TFAM gene expression in all intestinal tissues (duodenum, jejunum, ileum, cecum, and colon) (Figure 9). GEG administration resulted in lower TFAM gene expression levels in all studied small intestines and large intestine tissues of T2DM rats (Figure 9). 

### 3.5. mRNA Expression of the Intestinal Mitochondrial Mitophagy Markers Assessment

We assessed the effects of GEG supplementation on two mitophagy-associated parameters, LC3B (an autophagy formation marker) and P62 (an autophagy degradation marker) in intestinal tissues. The DM rats had increased LC3B gene expression levels in all intestinal tissues except for the colon (no significant changes) (Figure 10). Supplementation of GEG into the diet alleviated the LC3B gene expression in the above tissues (Figure 10). Similar to the findings of LC3B, the results of P62 analysis show that T2DM induced the gene expression of P62 in all studied intestinal tissues (Figure 11). GEG supplementation suppressed such increased P62 gene expression levels in these tissues in T2DM rats (Figure 11). 

Figure 12 shows the effects of GEG supplementation on gene expression of PINK1, a mitochondrial kinase, in the intestinal tissues of DM rats. Compared with non-DM rats (the control group), the DM mice had increased PINK1 gene expressions in the duodenum, ileum, cecum, and colon (Figure 12A). GEG supplementation resulted in decreased PINK1 gene expression levels in the duodenum, jejunum, cecum, and colon (Figure 12A) of DM rats. The IHC results of PINK1 protein expression confirmed the findings observed in the PINK1 mRNA gene expression in the colon (Figure 12B).

### 3.6. mRNA Expression of Intestinal Oxidative Stress and Inflammation Markers Assessment

Figure 13 shows the effects of GEG supplementation on SOD1 (a free radical threatening enzyme that offers protection against oxidative stress) gene expression in the intestinal tissues of DM rats. The DM rats had decreased SOD1 gene expression in the duodenum, cecum, and colon compared to the control group (Figure 13). Supplementation of GEG led to increased levels of SOD1 gene expression in the duodenum, ileum, cecum, and colon (Figure 13).

We also evaluated the effects of GEG supplementation on the inflammatory marker NF-κB gene expression in DM intestinal tissues (Figure 14). Relative to the control group, the DM group had increased NF-κB gene expression levels in the duodenum, jejunum, cecum, and colon; while GEG supplementation suppressed NF-κB gene expression in the above tissues, except for the ileum (Figure 14). 

## 4. Discussion

In the present study, the HFD+STZ model of T2DM was successfully employed to investigate the impact of GEG, supplemented in the diet for 6 weeks, on intestinal health in the DM rats. A link between an impaired intestinal barrier and the pathogenesis of T2DM has been previously reported [22]. In the current study, compared to the Control rats, the DM rats show decreased intestinal barrier function, as demonstrated by decreased gene and protein (colon) expression of Claudin-3. Our findings are consistent with the previous studies using HFD-induced prediabetic mice [22] and db/db diabetic mice [23,24]. Nascimento et al. reported that the tight junction-mediated barrier in intestinal tissues, such as the duodenum, jejunum, ileum and colon epithelia, was significantly weakened in prediabetic mice [22]. Ginger and its bioactive components have been shown to improve gastrointestinal health [25,26]. The current study is the first study to demonstrate the potent effects of dietary GEG supplementation on improving intestinal barrier function, as shown by increased Claudin-3 gene expression and immunofluorescence images in the colon of T2DM male rats. Our findings demonstrate that GEG’s effect on Claudin-3 in intestinal tissues is restricted to the colon and ileum, likely because the colon and ileum have the longest intestinal transit times of the gastrointestinal system [27]. 

To maintain mitochondrial and cellular functions, mitochondria are involved in active and dynamic processes, such as mitochondrial fusion, fission, biogenesis, and mitophagy. Mitochondria homeostasis is controlled by fusion and fission. Both mitochondrial fusion and fission are vital processes for repairing damaged components, allowing the exchange of material between damaged and non-damaged mitochondria via fusion, or segregation of damaged components via fission [28,29]. Mitochondria are one of the main sources of ROS and the major site of energy ATP production. Alterations to this balance (between fusion and fission) can involve oxidative stress, such as mitochondrial dysfunction or various metabolic alterations, eventually contributing to the development of mitochondria-related diseases, such as insulin resistance and T2DM [2]. In obesity and T2DM, impaired mitochondrial function (mitochondrial dysfunction), lowered rates of oxidative phosphorylation (ATP production), and excessive ROS production have been reported [30]. When levels of glucose are elevated (hyperglycemia) in obesity and T2DM, mitochondria enhance ROS production and induce oxidative stress, reducing ATP production and resulting in tissue damage [31]. The impaired mitochondrial function may lead to organ dysfunction, including in the intestines. In the present study, the changes in MFN1 and FIS1 in our HFD+STZ animals are consistent with previous studies on hyperglycemia-induced mitochondrial fission and fragmentation [32,33]. On the other hand, a ratio between MFN1 and FIS1 would further explain how GEG administration affects mitochondrial biogenesis. For example, in the present study, the ratios of MFN1/FIS1 in the colon for the control, DM, and GEG group are 0.5721 ± 0.021, 0.5207 ± 0.0249, and 0.5864 ± 0.023, respectively (*p* < 0.05). Compared to the DM group, although GEG group had decreased MFN1 and FIS1 gene expression, the GEG group had the greater MFN1/FIS1 ratio, suggesting GEG supplementation favors fusion over fission resulting in improved mitochondrial biogenesis production in the colon of diabetic rats.

PGC-1α is an important transcription coactivator regulating cellular energy metabolism. Mitochondrial dysfunction produces excessive ROS, eventually leading to oxidative stress. PGC-1α improves the balance between ROS production and its detoxification during inflammation by regulating key antioxidant gene expression [34]. The expression and the activity of PGC-1α are regulated by various cytokines, transcription factors, and other external stimuli via multiple intracellular signaling pathways [35]. Accumulating evidence indicates that PGC-1α is involved in the glucose homeostasis/regulation of T2DM in a variety of organs, such as the liver, muscle, pancreas, adipose tissue, kidney, and brain [35,36]. In the present study, our finding that the HFD+STZ rats had higher PGC-1α gene expression levels in intestinal tissues is also reported by others [36,37]. Hancock et al. reported that HFD-induced skeletal muscle mitochondria had increased mitochondrial DNA and proteins with a mechanism dependent on peroxisome proliferator-activated receptor activation and upregulation of PGC-1α [37]. PGC-1α promotes glucose production in the liver and inhibits insulin secretion by β cells, changes that promote T2DM [36]. The elevation of mitochondrial protein PGC-1α in our DM animals is likely due to increased fatty acid oxidation capacity in oxidative metabolism [38,39]. In the current work, we found that GEG supplementation suppressed PGC-1α gene expression in intestinal tissues of DM rats, indicating GEG’s involvement in antioxidative mechanisms. However, such effects on PGC-1α gene expression in intestines were not consistent with Li’s study using myocytes [15]. Li et al. showed ginger increased PGC-1α mRNA expression in myocytes of diabetic rats [15]. The discrepancy between findings may be due to the differences in targeted tissues (in vivo intestines in ours vs. in vitro myotubes in Li’s), delivery of GEG (in vivo diet in ours vs. in vitro culture media in Li’s), and length of GEG treatment (in vivo 6 weeks in ours vs. in vitro 10 min in Li’s). 

We also studied how GEG supplementation affects TFAM in the intestinal tissues of DM rats. The role of TFAM is to determine the abundance of the mitochondrial genome by regulating the packaging, stability, and replication [40]. The expression of TFAM is regulated by the nuclear respiratory factors 1 and 2 (NRF-1 and NRF-2), which promote mitochondrial biogenesis by inducing TFAM-dependent mitochondrial DNA replication and transcription [41]. The activation of NRF-1 and NRF-2, and the subsequent induction of TFAM, are governed by the master regulator of mitochondrial metabolism and biogenesis: PGC-1α [41]. In T2DM, the role of NRF-2 is to protect pancreatic β-cells against various insults (oxidative stress and inflammation), thereby maintaining glucose homeostasis and increasing insulin sensitivity [42]. In the current study, our findings that the DM rats had upregulated TFAM gene expression in all observed intestinal tissues are accompanied with the elevation of PGC-1α mRNA expression levels within intestines [41]. However, neither DM nor GEG affected NRF-2 mRNA expression levels (data not shown). The observation that the elevated TFAM gene expression levels in intestines were suppressed by GEG supplementation in the HFD+STZ rats is different from previous studies in 3T3-L1 adipocytes [43] and liver HepG2 cells [44]. Wang et al. demonstrated that 6-gingerol, the bioactive compound in ginger extract, greatly increased mitochondrial energy metabolism by increasing mitochondrial biogenesis markers, such as TFAM and NRF-1, in adipocytes [43]. Deng et al. showed that ginger extract promoted mitochondrial biogenesis via activation of AMPK-PGC-1α signaling pathways, as shown in increased PGC-1α and TFAM expression in HepG2 cells [44]. We noted that our studied animals are hyperglycemic, which are different from Wang’s and Deng’s cell studies. We suspect that this physiological difference may influence GEG treatment compared to animals with normal cell cultures. 

Mitophagy is a subtype of autophagy that selectively removes damaged mitochondria acting as a protective mechanism against oxidative stress in cells. Mitophagy can mitigate aging and damage to mitochondria, which is important for controlling mitochondrial quality [45]. LC3B is the marker for autophagosome formation, and p62 is the marker of autophagosome degradation [45]. The PINK1/Parkin-mediated pathway is one of the most mature mitophagy pathways in mammals [45]. Dysregulation of mitophagy is involved in the pathogenesis of a variety of metabolic and age-related diseases, including T2DM. Under the T2DM condition, a mitochondrion suffers depolarization and interrupts normal proteolytic processing of PINK1, resulting in an accumulation of PINK1 in the mitochondrion and phosphorylation of ubiquitin and Parkin. Then, Parkin mediates the ubiquitination of the outer mitochondrial membrane for binding p62 and autophagosome LC3 protein, eventually leading to the induction of mitophagy [2]. The observation that the HFD+STZ rats (the DM group) had significantly increased the gene expression of LC3B, P62, and PINK1 in the intestinal tissues suggests enhanced mitophagy via a PINK1/Parkin-mediated pathway in DM rats. Our findings that the DM rats increased intestinal mitophagy corroborates previous studies in diabetic animals [41,46,47]. Sergi et al. reported increased nutrient (i.e., HFD) supply generates an increase in mitochondrial ROS production, which can directly induce insulin resistance and elicit oxidative damage to mitochondrial DNA, proteins, and lipids, instigating the removal of damaged mitochondria via mitophagy [41]. Pontrelli et al. reported that mitophagy is dysregulated under diabetic conditions, as shown by increased LC3 and P62 expression in the renal tubular cells of patients with diabetic nephropathy [46]. Xiang et al. demonstrated that hyperglycemia induced mitochondrial dysfunction, shown by increased LC3B, PINK1, and Parkin in diabetic submandibular glands [47]. The autophagic process is the sole known mechanism for mitochondrial turnover and dysfunction of autophagy may lead to mitochondrial dysfunction and oxidative/nitrative stress [48]. In order to further elucidate how GEG supplementation affects autophagy, we calculated the ratio of LC3B and P62 gene expression in studied rats. For instance, the ratios of LC3B/P62 in the colon for the control, DM, and GEG group are 1.726 ± 0.1511, 1.328 ± 0.1645, 1.912 ± 0.1590, respectively (*p* < 0.05). Intriguingly, relative to the DM group, the GEG group had greater autophagy, as shown by an increased LC3B/P62 ratio, suggesting increased mitochondrial turnover. 

Hyperglycemia can increase ROS and reduce mitochondrial biogenesis, resulting in inflammation, tissue damage, and mitochondrial dysfunction [47,49]. Excessive ROS contributes to mitochondrial dysfunction and mitophagy in diabetic rats [47]. In addition, the mitochondrial dysfunction leads to a reduction in β-oxidation and ATP production, and possibly a further increase in ROS, resulting in insulin resistance and diabetes [47,49]. Hyperglycemia and dysfunctional mitochondria play a major role in this vicious cycle that causes inflammation, insulin resistance, and diabetes. Kabra et al. reported that increased fission and fragmentation of mitochondria was linked to hyperglycemia-induced overproduction of ROS and insulin secretion in mouse and human islets [33]. Therefore, to prevent mitochondrial dysfunction observed in T2DM, mitochondrial quality must be well regulated and maintained through mitochondrial fusion, fission, and mitophagy. In the present study, besides GEG’s balancing mitochondrial fusion, fission, and mitophagy, we also assessed two oxidative stress-related biomarkers, namely an antioxidant factor (SOD1) and an inflammation factor (NF-κB), within intestinal tissues. Our findings that the DM rats expressed lower SOD1 gene expression levels and higher NF-κB gene expression levels demonstrate the excessive ROS production within intestinal tissues [47]. GEG supplementation reverses the gene expression levels of SOD1 and NF-κB observed in the DM rats. Our results, supporting GEG’s anti-oxidative and anti-inflammatory effects, agree with Shanmugam et al. [50] and Ramudu et al. [51]. Shanmugam et al. showed that oral administration of ginger exhibits a neuroprotective effect by accelerating brain anti-oxidant defense mechanisms (an increase in SOD, catalase, glutathione peroxidase, glutathione reductase, as well as reduced glutathione) and down-regulating malondialdehyde levels in the diabetic rats [50]. Ramudu et al. also reported that ginger could lower the blood glucose levels as well as decrease activities of intra- and extra-mitochondrial enzymes in diabetic rats, namely lactate dehydrogenase (an index of oxidative stress activity). Such impacts from GEG on SOD1 and NF-κB would explain its beneficial effects on intestinal health through the reduction of mitochondrial dysfunction and the mitigation of mitophagy initiation. 

In this study, for GTT the rats were injected glucose. The insulin produced by the pancreas in the GEG group was able to respond and lowered glucose levels significantly when compared to the DM group, suggesting that the increased pancreatic insulin was able to function in GEG group. For ITT, the rats were injected with insulin, however, the GEG group did not respond to ITT, suggesting that GEG rats have insulin resistance preventing a response to the injected insulin. Furthermore, we demonstrated a significant increase in total pancreas insulin concentration in the GEG-supplemented rats compared to the diabetic non-GEG-supplemented rats. Analysis of the pancreas by immunostaining for insulin also revealed more insulin-producing beta cells in the GEG-supplemented DM rats, as compared to the DM rats. This phenomenon could be explained by either decreased beta cell death or increased beta cell proliferation. However, the analyses of tissue sections for apoptosis (by TUNEL) and proliferation (by Ki67 immunostaining) did not detect any apoptotic or proliferating islet cells (data not shown). Hence, future analysis of tissues collected closer to the time of STZ treatment is warranted to confirm the effects of GEG on pancreatic islet beta cell survival or proliferation. Interestingly, we found that the alpha cell localization was disrupted in the DM rats, probably due to the alpha cells filling in the space formerly occupied by beta cells. The alpha cells were also intermixed with the beta cells in the GEG group (rather than being on the outer edge, as in the control group). Our findings suggest that GEG supplementation did not prevent STZ-induced beta cell death. Instead, GEG supplementation must have increased the number of beta cells after STZ administration, since decreased beta cell death would have resulted in normal alpha cell distribution. In other words, the alpha cells would have remained on the outer edge of the islets and not moved into the space evacuated by the beta cells. Collectively, this study suggests that GEG supplementation resulted in improved pancreatic insulin production, most likely by inducing beta cell proliferation in the DM rats. 

## 5. Conclusions

In the current study, we demonstrated that GEG supplementation improved glucose homeostasis, as shown by improved GTT and increased pancreas insulin staining in DM rats. Such improvement in glucose homeostasis may be, in part, due to improved GI health, as shown by (i) the improvement in intestinal barrier function and mitochondrial function (biogenesis and autophagy), and (ii) the reduction in inflammation and oxidative stress in diabetic rats. The ginger extract used in this study is a very rough extract (a combination of all ginger bioactive compounds). Future study is warranted to further extract the ginger extract in order to clarify the main active anti-diabetic components in management of T2DM. 

## Figures and Tables

**Figure 1 nutrients-14-04384-f001:**
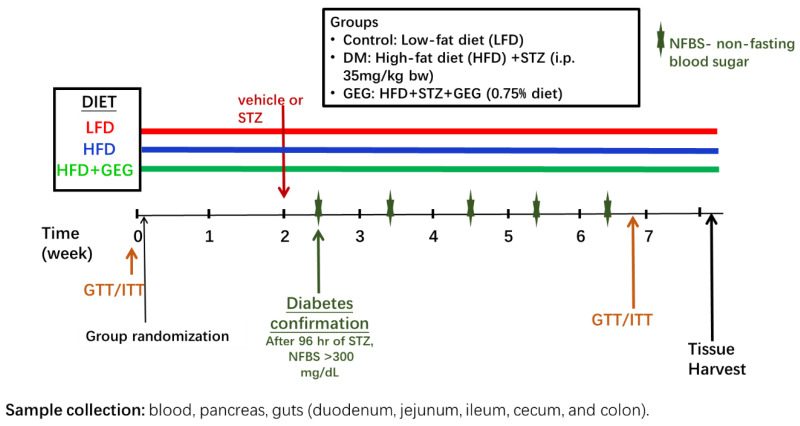
Experimental design describing the three experimental groups, diet assigned to each group, GTT/ITT tests, STZ/vehicle administration, non-fasting blood sugar assessment (NFBS), and tissue collection.

**Figure 2 nutrients-14-04384-f002:**
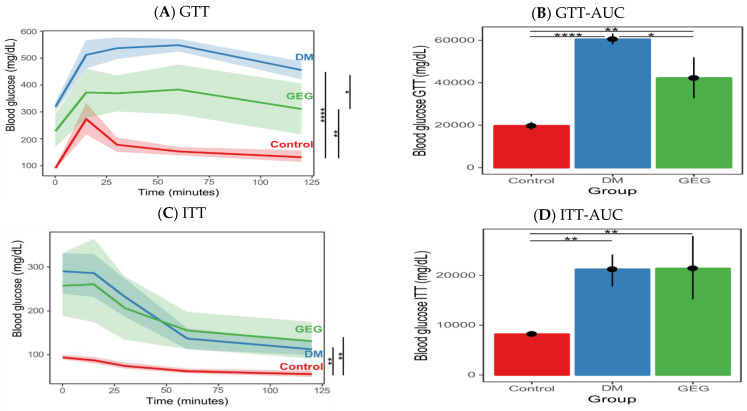
Effect of gingerol-enriched ginger (GEG) on blood glucose during ipGTT (**A**), ipGTT AUC (**B**), blood glucose during ipITT (**C**), and ipITT AUC (**D**). Data is presented as mean with error bars or ribbons indicating 90% confidence interval. One-way ANOVA was used, followed by Tukey’s HSD in R environment. *n* = 6–8 per group. * indicates *p* < 0.05, ** *p* < 0.01, and **** *p* < 0.0001.

**Figure 3 nutrients-14-04384-f003:**
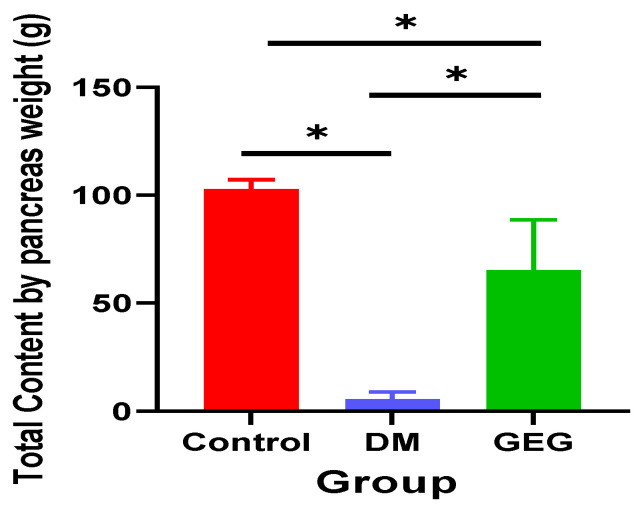
Effect of gingerol-enriched ginger (GEG) on total pancreatic insulin content. Data is expressed as mean ± SEM. *n* = 6–8 per group. Data was analyzed by one-way ANOVA followed by a post hoc Tukey’s test. * *p* < 0.05.

**Figure 4 nutrients-14-04384-f004:**
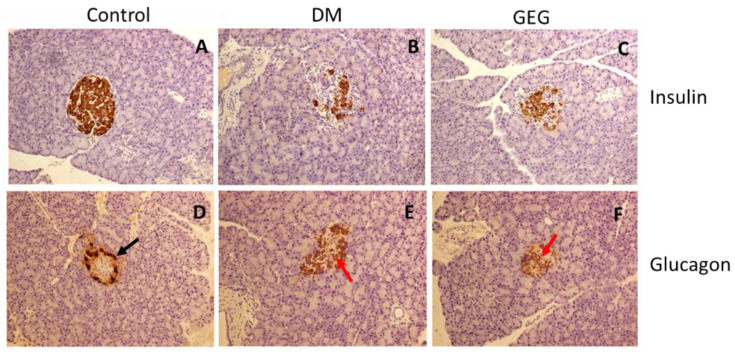
Effect of gingerol-enriched ginger (GEG) on immunohistochemical analysis of pancreatic tissue. Pancreatic tissue sections collected from the Control group (**A**,**D**), DM group (**B**,**E**), and GEG group (**C**,**F**) were immunostained for insulin (**A**–**C**) or glucagon (**D**–**F**) and counterstained with hematoxylin. The black arrow in (**D**) indicates alpha cells in the normal orientation along the outer edge of the islet, while the red arrows in (**E**,**F**) indicate alpha cells localized to the center of the islet. *n* = 6–8 per group. IHC magnification 20×.

**Figure 5 nutrients-14-04384-f005:**
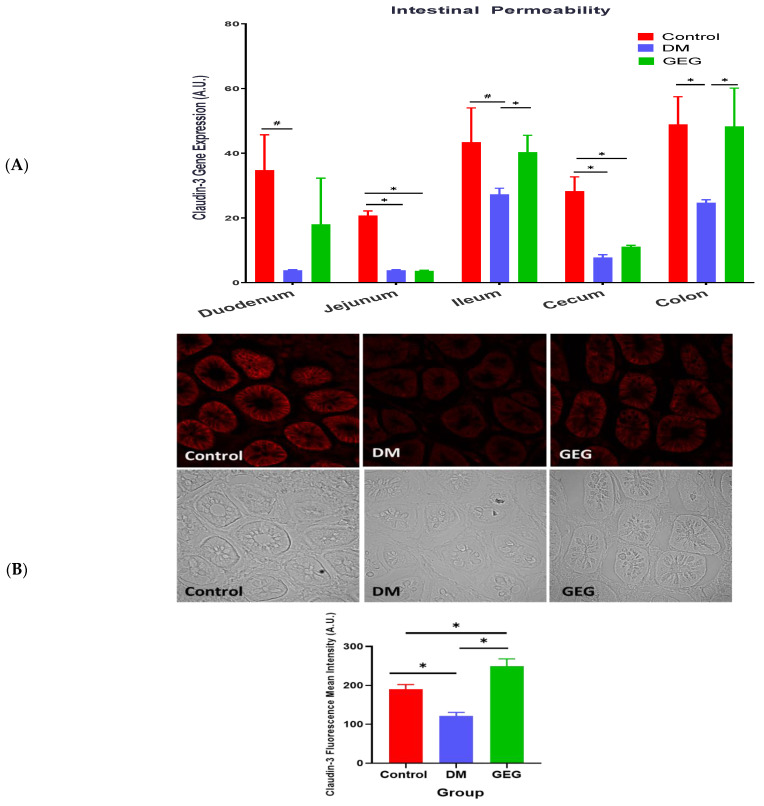
Effect of gingerol-enriched ginger (GEG) on mRNA gene expression of Claudin-3 in duodenum, jejunum, ileum, cecum, and colon of rats (**A**). Colon tissue sections were immunostained for Claudin-3 along with quantification of Claudin-3 fluorescence intensity (**B**). Data is expressed as mean ± SEM. *n* = 6–8 per group. Data was analyzed by one-way ANOVA followed by a post hoc Tukey’s test. * *p* < 0.05. # 0.05 < *p* < 0.1. IHC magnification 60×.

**Figure 6 nutrients-14-04384-f006:**
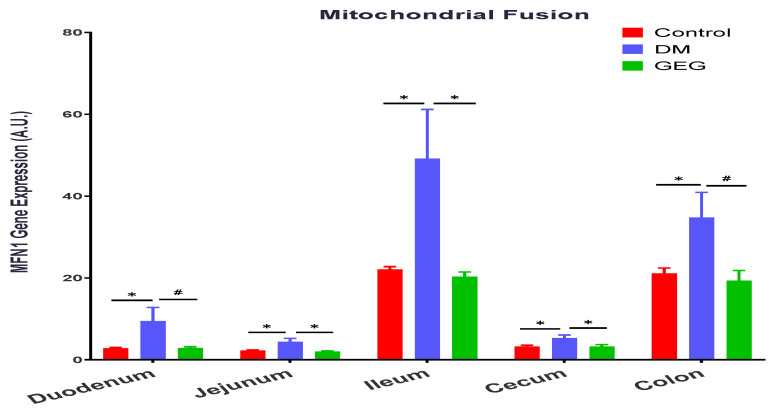
Effect of gingerol-enriched ginger (GEG) on mRNA gene expression of MFN1 3 in duodenum, jejunum, ileum, cecum, and colon of rats. Data is expressed as mean ± SEM. *n* = 6–8 per group. Data was analyzed by one-way ANOVA followed by a post hoc Tukey’s test. * *p* < 0.05. # 0.05 < *p* < 0.1.

**Figure 7 nutrients-14-04384-f007:**
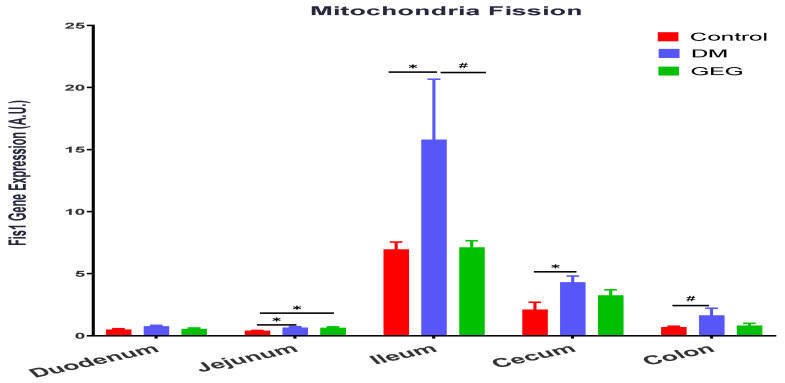
Effect of gingerol-enriched ginger (GEG) on mRNA gene expression of FIS1 in duodenum, jejunum, ileum, cecum, and colon of rats. Data is expressed as mean ± SEM. *n* = 6–8 per group. Data was analyzed by one-way ANOVA followed by a post hoc Tukey’s test. * *p* < 0.05. # 0.05 < *p* < 0.1.

**Figure 8 nutrients-14-04384-f008:**
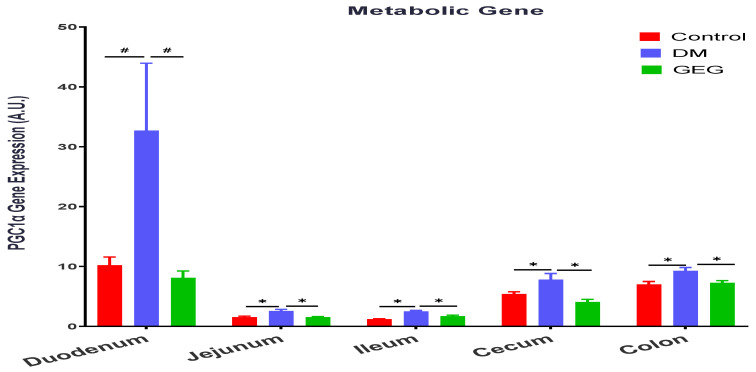
Effect of gingerol-enriched ginger (GEG) on mRNA gene expression of PGC1α in duodenum, jejunum, ileum, cecum, and colon of rats. Data is expressed as mean ± SEM. *n* = 6–8 per group. Data was analyzed by one-way ANOVA followed by a post hoc Tukey’s test. * *p* < 0.05. # 0.05 < *p* < 0.1.

**Figure 9 nutrients-14-04384-f009:**
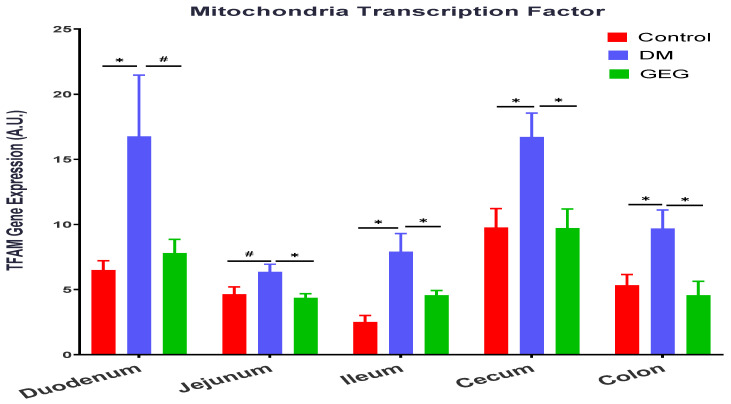
Effect of gingerol-enriched ginger (GEG) on mRNA gene expression of TFAM in duodenum, jejunum, ileum, cecum, and colon of rats. Data is expressed as mean ± SEM. *n* = 6–8 per group. Data was analyzed by one-way ANOVA followed by a post hoc Tukey’s test. * *p* < 0.05. # 0.05 < *p* < 0.1.

**Figure 10 nutrients-14-04384-f010:**
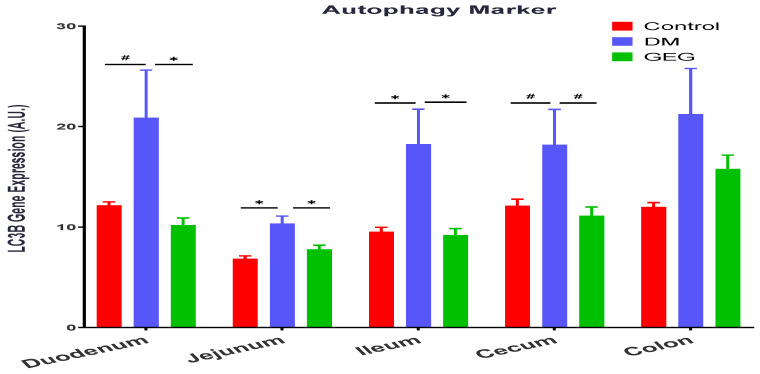
Effect of gingerol-enriched ginger (GEG) on mRNA gene expression of LC3B in duodenum, jejunum, ileum, cecum, and colon of rats. Data is expressed as mean ±SEM. *n* = 6–8 per group. Data was analyzed by one-way ANOVA followed by a *post hoc* Tukey’s test. * *p* < 0.05. # 0.05 < *p* < 0.1.

**Figure 11 nutrients-14-04384-f011:**
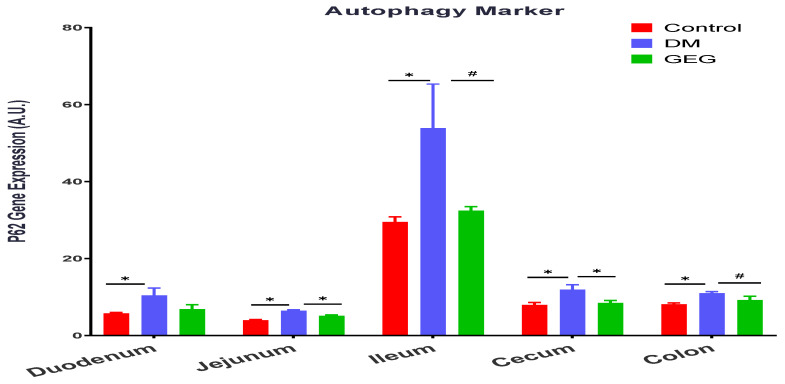
Effect of gingerol-enriched ginger (GEG) on mRNA gene expression of P62 in duodenum, jejunum, ileum, cecum, and colon of rats. Data is expressed as mean ± SEM. *n* = 6–8 per group. Data was analyzed by one-way ANOVA followed by a post hoc Tukey’s test. * *p* < 0.05. # 0.05 < *p* < 0.1.

**Figure 12 nutrients-14-04384-f012:**
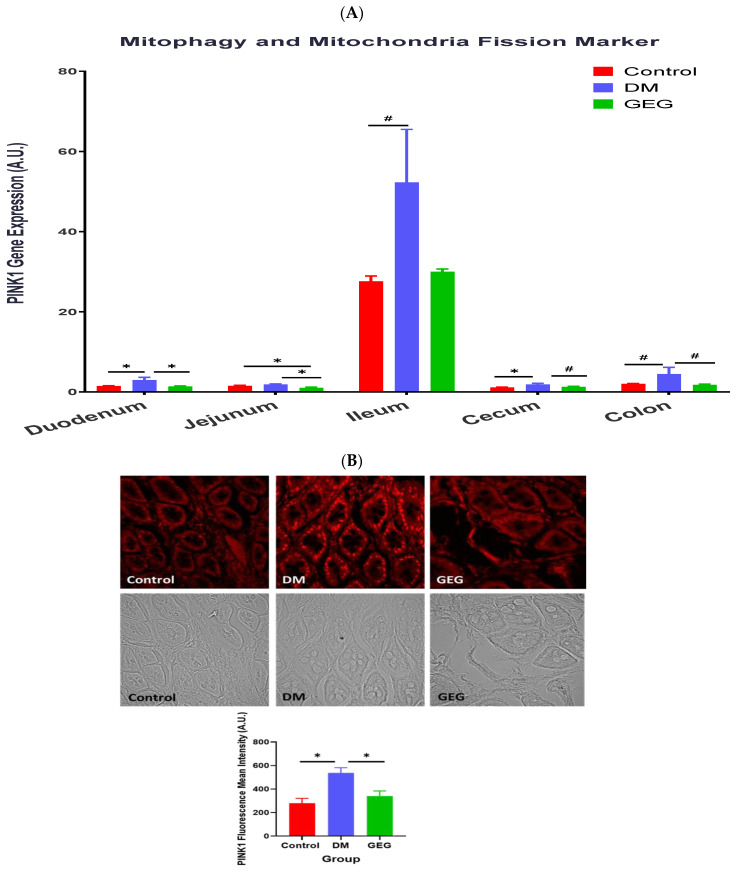
Effect of gingerol-enriched ginger (GEG) on mRNA gene expression of PINK1 in duodenum, jejunum, ileum, cecum, and colon of rats (**A**). Colon tissue sections were immunostained for PINK1 along with quantification of PINK1 fluorescence intensity (**B**). Data is expressed as mean ± S EM. *n* = 6–8 per group. Data was analyzed by one-way ANOVA followed by a post hoc Tukey’s test. * *p* < 0.05. # 0.05 < *p* < 0.1. IHC magnification 60×.

**Figure 13 nutrients-14-04384-f013:**
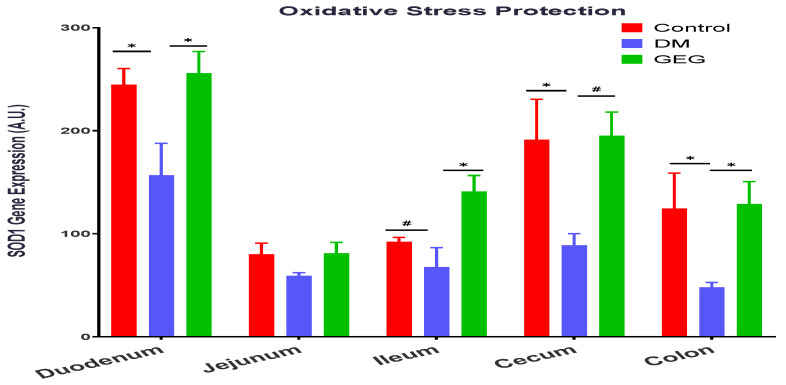
Effect of gingerol-enriched ginger (GEG) on mRNA gene expression of SOD1 in duodenum, jejunum, ileum, cecum, and colon of rats. *n* = 6–8 per group. Data is expressed as mean ± SEM. *n* = 6–8 per group. Data was analyzed by one-way ANOVA followed by a post hoc Tukey’s test. * *p* < 0.05. # 0.05 < *p* < 0.1.

**Figure 14 nutrients-14-04384-f014:**
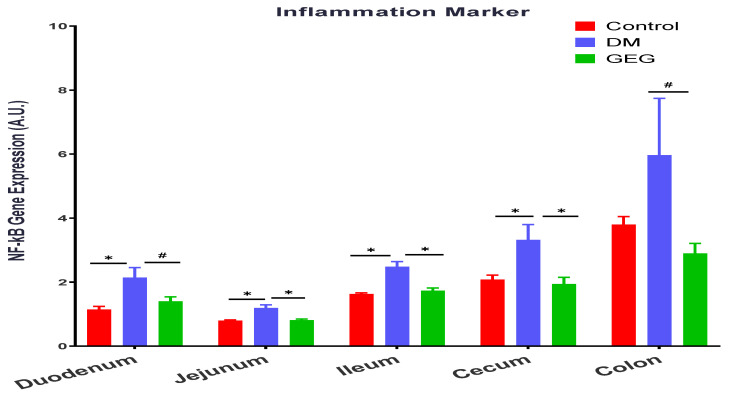
Effect of gingerol-enriched ginger (GEG) on mRNA gene expression of NF-kB in duodenum, jejunum, ileum, cecum, and colon of rats. Data is expressed as mean ± SEM. *n* = 6–8 per group. Data was analyzed by one-way ANOVA followed by a post hoc Tukey’s test. * *p* < 0.05. # 0.05 < *p* < 0.1.

**Table 1 nutrients-14-04384-t001:** List of primers for mRNA.

Gene	Forward	Reverse
Claudin-3	5′-CCC AGC CTA CGG AGT TAC CC-3′	5′-TGC CGA TGA ATG CCG AAA CG-3′
MFN1	5′-AGC TCG CTG TCA TTG GGG AG-3′	5′-TCC CTC CAC ACT CAG GAA GC-3′
FIS1	5′-CTG CGG TGC AGG ATG AAA GAC-3′	5′-GGC GTA TTC AAA CTG CGT GCT-3′
PGC-1α	5′-CAG GAG CTG GAT GGC TTG GG-3′	5′-GGG CAA AGA GGC TGG TCC T-3′
TFAM	5′ -GCT TCC AGG GGG CTA AGG ATG-3′	5′-TCG CCC AAC TTC AGC CAT TT-3′
P62	5′-CTG AGT CGG CTT CTG CTC CA-3′	5′-GCG GCT TCT CTT CCC TCC AT-3′
LC3B	5′-CAT GCC GTC CGA GAA GAC CT-3′	5′-CCG GAT GAG CCG GAC ATC TT-3′
PINK1	5′ -TCG GCC TGT CAG GAG ATC CA-3′	5′-CAT TGC AGC CCT TGC CGA TG-3′
SOD1	5′-AGG GCG TCA TTC ACT TCG AG-3′	5′-ACA TGC CTC TCT TCA TCC GCT-3′
NF-kB	5′-CCT CCA CCC CGA CGT ATT GC-3′	5′-GCC AAG GCC TGG TTT GAG AT-3′
β-actin	5′-ACA ACC TTC TTG CAG CTC CTC C-3′	5′-TGA CCC ATA CCC ACC ATC ACA-3′

Abbreviations: MFN1, mitofusin 1; FIS1, fission 1 protein; PGC-1α, peroxisome proliferator-activated receptor gamma coactivator 1 alpha; TFAM, transcription factor A, mitochondrial; PINK1, (PTEN)-induced putative kinase 1; P62, ubiquitin-binding protein 62; LC3B, microtubule-associated protein 1 A/1B-light chain 3; SOD1, superoxide dismutase 1; NF-kB, nuclear factor kappa-light-chain-enhancer of activated B cells.

## Data Availability

Not applicable.

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
