# Peer review of "Ginger Root Extract Improves GI Health in Diabetic Rats by Improving Intestinal Integrity and Mitochondrial Function"

_nutrients, 2022, doi:10.3390/nu14204384_

Round 1

Reviewer 1 Report

The authors used gingerol-enriched ginger (GEG) extract to treat diabetes rats fed with high fat diet, the therapeutic effects of GEG on diabetes rats were mainly discussed from three aspects: maintaining intestinal homeostasis, improving mitochondrial dysfunction, and inhibiting inflammatory response and oxidative stress. It is concluded that GEG has the potential to improve blood glucose homeostasis, which potential is closely related to intestinal health.

This study is of great significance and value for the development of beneficial bioactive components with anti-diabetes. However, this paper may be accepted in Nutrients after major revision.

1.     The departments to which the authors belong are not specific.

2.     Title: I think this title is too long and has too many elements. It should be simplified with keywords.

3.     Abstract: In this part, the background, methods, results and conclusions of the experiment were briefly described, however, there are still several problems: Firstly, in the “Background”, before introducing the main content of this study, it should explain the existing research foundation of the relationship between GEG and diabetes briefly.

4.     Secondly, in the “Methods”, please leave a blank space between numbers and units.

5.     Thirdly, the application prospect of this work needs to be added at the end of the “conclusion”.

6.     Keywords: It is more appropriate to change “bioactive compound” to “ginger root extract”.

7.     Introduction: At the beginning of the first paragraph, I think “characterized by” should be changed to “mainly characterized by”, the symptoms of type 2 diabetes are not only hyperglycemia and insulin resistance.

8.     The coherence of the four sentences in the first paragraph of the “Introduction” is not strong, I suggest that it be changed to: “Type 2 diabetes mellitus (T2DM), mainly characterized by hyperglycemia and insulin resistance, is the fastest-growing metabolic disease in the world. Accumulating evidence has highlighted a strong correlation between T2DM, intestinal barrier dysfunction, oxidative stress, and mitochondrial dysfunction. Among them, hyperglycemia, insulin resistance and insulin damage caused by inflammatory cytokines and oxidative stress are the main reasons”.

9.     The first line of the second paragraph of “Introduction”, I think “proper” should be replaced by “normal”, “proper” is more likely to mean correct or appropriate.

10.  After introducing the relationship between intestinal barrier integrity and diabetes (the second paragraph of “Introduction”), I suggest that you can simply mention the positive effects of food active substances on both.

11.  About mitochondrial function, I have doubts about “principle of the T2DM phenotype” mentioned at the end of the third paragraph of the introduction. What is its specific origin and meaning? Please indicate the reference.

12.  It is the same sentence as point 6, the subject of the attributive clause guided by "that" is not clear. I suggest combining two short sentences directly, for example, “such as dietary bioactive compounds can improve mitochondrial function to treat T2DM and associated comorbidities”.

13.  In the fourth paragraph of the “Introduction”, the subject and logical relationship of the first sentence and the second sentence are identical, I suggest combining them with phrases or prepositions.

14.  In the fourth paragraph of the “Introduction”, I suggest to delete “Among different bioactive compounds”, then directly explain the effect of ginger extract according to existing reports.

15.  In the fourth paragraph of the “Introduction”, whether enough clinical studies have confirmed that GEG can indeed be used as a potential candidate for management of T2DM, and whether to consider adding words like "maybe".

16.  In the fourth paragraph of the “Introduction”, since “the intestinal tight junction protein” is to be used as an indicator to observe intestinal homeostasis, why is it not mentioned at all in the second paragraph of “Introduction”?

17.  Please indicate that the large and small intestine samples are from diabetes rats.

18.  I don't think "on intestines" is appropriate, maybe you can change into “The anti-diabetes effect of GEG through the intestine is partly mediated via suppression of intestinal oxidative stress and inflammation”.

19.  What is the source of GEG?

20.  The title of the chart is not indicated.

21.  I think the section title of the “Results” can make the keywords specific. It would be better to write the "results" in a specific and complete sentence.

22.  What exactly does the "baseline" mentioned all the time refer to?

23.  What does “n = 6-8 per group” mean in 3.2? Are the sample numbers of the three groups inconsistent?

24.  The coordinate axis of the bar chart is not very beautiful. I suggest adjusting and making the format as uniform as possible.

25.  In the Figure 3C and 3F, I don't think “alpha cells were also located throughout the islet intermixed with the beta cells” is apparent.

26.  Regarding intestinal homeostasis (3.3), can only Claudin-3 prove that there is a problem with the intestinal barrier?

27.  In figure 4B, “GRE” should be changed to “GEG”, Figure 11B shows the same problem.

28.  I think you can take the graph of Claudin-3 protein expression in colon as figure 4C alone.

29.  “Endotoxemia” (in the second sentence of the first paragraph of the Discussion) can be deleted, which has nothing to do with the content of the article.

30.  The “mitochondrial turnover rate” suddenly introduced is very puzzling. I think it should be explained simply.

31.  Can you supplement or quote data of PGC-1α gene expression levels?

32.  It links mitochondrial function with oxidative stress and type 2 diabetes using PCG-1α, but it is too long and can be simplified. (The third paragraph of the discussion)

33.  The text font and size of the figure notes in the paper are not uniform, the figures need to be re-integrated.

34.  I think there should be a blank between all numbers and units.

35.  “Thus” appears too frequently in the article. You can replace it with “consequently”, “for this reason”, “hence” and so on.

36.  In the conclusions, the application, significance and research prospect of GEG should be pointed out.

Author Response

September 30, 2022

Re:  Ginger root extract improves intestinal health in diabetic rats via improving intestinal integrity and mitochondrial function and suppressing oxidative stress and inflammation

Dear Editor-In-Chief:

Thank you so much for the thorough review of our manuscript. We now submit our revision for your consideration for publication.  We have carefully addressed comments made by reviewers.  Please see our response below with changes in highlighted yellow in the revised manuscript attached.

We believe that we have responded to all reviewers’ concerns and comments, and look forward to your favorable decision. Please direct all the communications to me.

Sincerely yours,

Chwan-Li (Leslie) Shen, PhD, CCRP (SoCRA)

Professor of Pathology, School of Medicine

Texas Tech University Health Sciences Center

Lubbock, TX 79430-8115, USA

Tel: 806-743-2815

EMail: [email protected]

Reviewer 2 Report

This study shows that ginger extract can effectively improve diabetic inflammation and intestinal function.

1. It is necessary to provide the gas chromatography-mass spectrometry analysis of ginger extract or cite previously published articles.

2. The ginger extract used in this study is a very rough extract, and the authors believe that the content of 6-gingerol is 18.7%, the content of 8-gingerol is 1.81%, and the content of 10-gingerol is 2.86%, the content of 6-shogaol was 3.09%, the content of 8-shogaol was 0.39%, and the content of 10-shogaol was 0.41%. However, these components together do not exceed 30%. It is suggested that the author further extract the ginger extract to clarify the main active components of anti-diabetic. Although the authors cite previous findings, however, the published paper is not convincing.

3. Lack of figure legends in the text of the paper.

4. Immunohistochemical analysis, specific to Figure 3, requires pictures under a low magnification microscope, which can make readers more convincing. Scanning electron microscopy is recommended to image the structure of the gut.

5. Figure 4 requires the author to provide a photo under the light microscope to confirm that this is the result in the intestinal tissue. Need to provide photos of different parts of the intestines.

6. The RT-PCR results are not convincing (Fig. 5-Fig. 13), especially the SD is larger. WB is recommended for verification.

Author Response

Reviewer #2:

Comments and Suggestions for Authors

This study shows that ginger extract can effectively improve diabetic inflammation and intestinal function.

  1. It is necessary to provide the gas chromatography-mass spectrometry analysis of ginger extract or cite previously published articles.

Response: Thanks for the comment. In this revision, we have included citation of previous paper (line 90).

  1. The ginger extract used in this study is a very rough extract, and the authors believe that the content of 6-gingerol is 18.7%, the content of 8-gingerol is 1.81%, and the content of 10-gingerol is 2.86%, the content of 6-shogaol was 3.09%, the content of 8-shogaol was 0.39%, and the content of 10-shogaol was 0.41%. However, these components together do not exceed 30%. It is suggested that the author further extract the ginger extract to clarify the main active components of anti-diabetic. Although the authors cite previous findings, however, the published paper is not convincing.

Response: This is a great suggestion to further extract the ginger extract in order to clarify the main active anti-diabetic components. In this revision, we have included such aspect as a possible future study in Conclusion (line 424-430).

  1. Lack of figure legends in the text of the paper.

Response: Thank you for the comments. In this revision, all figure legends are included.

  1. Immunohistochemical analysis, specific to Figure 3, requires pictures under a low magnification microscope, which can make readers more convincing. Scanning electron microscopy is recommended to image the structure of the gut.

Response: Thanks for the suggestion. The implication of IHC is to confirm the findings of mRNA gene expression in colon tissue, instead of showing the image of gut structure. Thus, the 60x magnification would serve such purpose by showing the location and the level of protein expression of the target protein in the colon.

  1. Figure 4 requires the author to provide a photo under the light microscope to confirm that this is the result in the intestinal tissue. Need to provide photos of different parts of the intestines.

Response: Thanks for the suggestion. In this revision, we have included bright field pictures in Figures 5 (#4 in the previous version) and 12, accordingly.

  1. The RT-PCR results are not convincing (Fig. 5-Fig. 13), especially the SD is larger. WB is recommended for verification.

Response: Thanks for the suggestion. We understand that some genes and some specific tissues (ex: ileum) showed more variability within the animals for some groups. Thus, we performed IHC for the colon staining to confirm the gene findings for some target proteins (Claudin-3 and PINK1). Due to the limitation of tissues, we were not able to perform western blots. For a good quality control of gene expression, we performed the gene expression in duplicate in every single sample on the same plate to minimize possible technical errors.

Reviewer 3 Report

In this study, the "anti-diabetic", mitochondrial and inflammatory-protective effects of a ginger root extract (GEG) are explored in a rat model. Potentially as bioactive compounds and mediated by oxidative stress pathways. These are data in animal models, I believe that it is a well-developed and comprehensible article, it provides the novelty of a nutritional product with anti-diabetic and intestinal barrier protective properties. However, I would like to make some comments.

- The title should include "in rats or in an animal model" and perhaps remove the part about "suppressing of oxidative stress and inflammation", since only SOD1 and NF-kB have been explored by qRT-PCR. 

- The introduction is well developed, and although extensive, it is understandable and well organized. 

- Again, the material and methods are very understandable. However, I would add a schematic figure with the study desing, when the STZ or GEG are supplied and at what times samples are collected. 

- In the statistical section, which R packages were used?, was RStudio used?, which versions?, Also, when describing the cut-off of the P-value, it is not necessary to describe each "*" since, once it exceeds the point set at 0.05, the authors already considered it significant. 

- On the other hand, why was the high fat diet necessary?, do you think that the effects of GEG could be comparable by treating the rats with STZ only?

- In the results section, although it is true that GEG improved glucose overload, however, it did not produce effects in the ITT and when determining the amount of insulin in the pancreas it was above the DM group, could this insulin be non-functional and that GEG did not improve it?

- This has to do with the organization of the islet, it does not seem that the GEG improves the organization.

- For the gene expressions, I suggest to include in the title of the figure the function of the protein (instead of MFN1, write mitochondrial fusion. Instead of FIS1, write mitochondrial fission. Leave MFN1 or FIS1 in Y axis). On the other hand, a ratio between MFN1/FIS1 or between LC3B/P62 could be applied, so that it can be determined whether fission or autophagy marker formation predominates. 

- It seems that GEG decreases the markers of mitochondrial biogenesis production, could DM rats have higher bioenergetics because their oxidative production is higher?. It is not clear to me if autophagy improves with GEG treatment (maybe a ratio could make it more understandable).

- I recommend that in the figure caption be included. In figures with only 3 bars (Fig. 2, for example), write the name of the groups on the X axis. Also, the caption of the image in Fig. 4A write "GRE".

- Finally, in the conclusion, supplementation with GEG did not improve ITT, I think this should be revised. Also, more details could be given, if GEG improved mitochondrial biogenesis or decreased fusion vs. fission, or what specific elements dimisnuted inflammation. 

Author Response

Reviewer #3.

Comments and Suggestions for Authors

In this study, the "anti-diabetic", mitochondrial and inflammatory-protective effects of a ginger root extract (GEG) are explored in a rat model. Potentially as bioactive compounds and mediated by oxidative stress pathways. These are data in animal models, I believe that it is a well-developed and comprehensible article, it provides the novelty of a nutritional product with anti-diabetic and intestinal barrier protective properties. However, I would like to make some comments.

Response: Thank you for your comments about our manuscript.

The title should include "in rats or in an animal model" and perhaps remove the part about "suppressing of oxidative stress and inflammation", since only SOD1 and NF-kB have been explored by qRT-PCR. 

Response: Thank you for the suggestion. In this revision, we have revised the title accordingly.

The introduction is well developed, and although extensive, it is understandable and well organized. 

Response: Thank you for your comments about our introduction.

Again, the material and methods are very understandable. However, I would add a schematic figure with the study design, when the STZ or GEG are supplied and at what times samples are collected.

Response: Thank you for your suggestion. In this revision, we have included a schematic figure with study design accordingly (Figure 1).

In the statistical section, which?, was R Studio used?, which versions?, Also, when describing the cut-off of the P-value, it is not necessary to describe each "*" since, once it exceeds the point set at 0.05, the authors already considered it significant. 

Response: In this revision, we have included the source/version of RStudio (line 159) and removed the description for *, **, and ***.    

On the other hand, why was the high fat diet necessary? do you think that the effects of GEG could be comparable by treating the rats with STZ only?

Response: A combination of high-fat diet-fed and low-dose STZ-treated rats has been widely used to study the effects of bioactive compounds or pharmacological screening in Type 2 diabetes (Srinivasan 2005, Zhang 2008, Khan 2012, Vatandoust 2018). Depending on the dose of STZ, the rats develop Type 1 diabetes phenotype with high dose(s) STZ or Type 2 diabetes phenotype with low dose(s) STZ and HFD.

Both HFD+STZ or STZ only have been used to study the effects of bioactive compounds on diabetes. Compared to STZ only, this HFD+STZ is more comparable to human T2DM. Nevertheless, we think the effects of GEG could be comparable by treating the rats with low dose STZ in animals.

Reference:

  • Srinivasan K, Viswanad B, Asrat L, Kaul CL, Ramarao P. Combination of high-fat diet-fed and low-dose streptozotocin-treated rat: a model for type 2 diabetes and pharmacological screening. Pharmacol Res. 2005 Oct;52(4):313-20. doi: 10.1016/j.phrs.2005.05.004. PMID: 15979893.
  • Zhang M, Lv XY, Li J, Xu ZG, Chen L. The characterization of high-fat diet and multiple low-dose streptozotocin induced type 2 diabetes rat model. Exp Diabetes Res. 2008;2008:704045. doi: 10.1155/2008/704045. Epub 2009 Jan 4. PMID: 19132099; PMCID: PMC2613511.
  • Khan HB, Vinayagam KS, Palanivelu S, Panchanadham S. Ameliorating effect of Semecarpus anacardium Linn. nut milk extract on altered glucose metabolism in high fat diet STZ induced type 2 diabetic rats. Asian Pac J Trop Med. 2012 Dec;5(12):956-61. doi: 10.1016/S1995-7645(12)60181-3. PMID: 23199713.
  • Vatandoust N, Rami F, Salehi AR, Khosravi S, Dashti G, Eslami G, Momenzadeh S, Salehi R. Novel High-Fat Diet Formulation and Streptozotocin Treatment for Induction of Prediabetes and Type 2 Diabetes in Rats. Adv Biomed Res. 2018 Jul 2;7:107. doi: 10.4103/abr.abr_8_17. PMID: 30069438; PMCID: PMC6050973.

In the results section, although it is true that GEG improved glucose overload, however, it did not produce effects in the ITT and when determining the amount of insulin in the pancreas it was above the DM group, could this insulin be non-functional and that GEG did not improve it?

Response: Thanks for the comments. For GTT, the rats were injected with glucose. The insulin produced by the pancreas in the GEG group was able to respond and lowered glucose levels significantly, when compared to the DM group, suggesting that the increased pancreatic insulin was able to function in GEG group. For ITT, the rats were injected with insulin, However, GEG group did not respond to ITT, suggesting that GEG rats have insulin resistance and they are not responding to the injected insulin. In this revision, we have included the above observation/explanation in the Discussion (line 404-4048).

This has to do with the organization of the islet, it does not seem that the GEG improves the organization.

Response: Thanks for the comments. It is true that the islet organization did not go back to normal rat islet organization in the GEG group. As GTT improved, the beta cells may be proliferating; islet organization may not play a role in this aspect in the GEG group. Furthermore, islets in humans and other species do not maintain this type of organization as alpha and beta cells can be intermixed. The importance of islet organization is outside the scope of the present study; however, this topic could be an interesting study in the future.

For the gene expressions, I suggest to include in the title of the figure the function of the protein (instead of MFN1, write mitochondrial fusion. Instead of FIS1, write mitochondrial fission. Leave MFN1 or FIS1 in Y axis).

Response: Thanks for the suggestion. We have included the titles and info on y-axis to specify the function of the protein in gene expression graphs accordingly.

On the other hand, a ratio between MFN1/FIS1 or between LC3B/P62 could be applied, so that it can be determined whether fission or autophagy marker formation predominates. 

Response: Thanks for the great suggestion. In this revision, we have included the ratios for MFN1/FIS1 (line 315-323) and LC3B/P62 (line 377-382) along with interpretation in the Discussion.

It seems that GEG decreases the markers of mitochondrial biogenesis production, could DM rats have higher bioenergetics because their oxidative production is higher? It is not clear to me if autophagy improves with GEG treatment (maybe a ratio could make it more understandable).

Response: Thanks for the great suggestion. In this revision, we have shown the increased ratios for MFN1/FIS1 (line 315-323) and LC3B/P62 (line 377-382) along with interpretation in the GEG group in the Discussion.

I recommend that in the figure caption be included. In figures with only 3 bars (Fig. 2, for example), write the name of the groups on the X axis. Also, the caption of the image in Fig. 4A write "GRE".

Response: Thanks for the suggestion. In this revision, the figure captions have been included.

Finally, in the conclusion, supplementation with GEG did not improve ITT, I think this should be revised. Also, more details could be given, if GEG improved mitochondrial biogenesis or decreased fusion vs. fission, or what specific elements dimisnuted inflammation. 

Response: Thanks for the comment. In this revision, we have revised Conclusion (line 424-430).

Round 2

Reviewer 1 Report

I think this revised version  can be accepted  .

Reviewer 2 Report

The revised version of the manuscript has made great improvements. However, as the top magazine in this field, I think these improvements are still insufficient.

1.      It is suggested that the author further extract the ginger extract to clarify the main active components of anti-diabetic. Although the authors cite previous findings, however, the published paper is not convincing.

2.      Immunohistochemical analysis, requires pictures under a low magnification microscope, which can make readers more convincing.

3.      Scanning electron microscopy is recommended to image the structure of the gut.

4.      The RT-PCR results are not convincing (Fig. 5-Fig. 13), especially the SD is larger. WB is recommended for verification.

These problems are critical and need to be solved. The author evades these questions.